# Transferrin receptor regulates malignancies and the stemness of hepatocellular carcinoma-derived cancer stem-like cells by affecting iron accumulation

Chong Xiao[1☯], Xi Fu[2☯], Yuting Wang[2], Hong Liu[2], Yifang Jiang[2], Ziyi Zhao[2]*, Fengming You[2]*

1 Chengdu University of Traditional Chinese Medicine, Chengdu, China, 2 Hospital of Chengdu University of Traditional Chinese Medicine, Chengdu, China

☯ These authors contributed equally to this work.
* zhaoziyi925@163.com (ZZ); youfengming@cdutcm.edu.cn (FY)

## Abstract

### Background

Iron metabolism is essential because it plays regulatory roles in various physiological and pathological processes. Disorders of iron metabolism balance are related to various cancers, including hepatocellular carcinoma. Cancer stem-like cells (CSCs) exert critical effects on chemotherapy failure, cancer metastasis, and subsequent disease recurrence and relapse. However, little is known about how iron metabolism affects liver CSCs. Here, we investigated the expression of transferrin receptor 1 (TFR1) and ferroportin (FPN), two iron importers, and an upstream regulator, iron regulatory protein 2 (IRP2), in liver hepatocellular carcinoma (LIHC) and related CSCs.

### Methods

The expression levels of TFR1, FPN and IRP2 were analysed using the GEPIA database. CSCs were derived from parental LIHC cells cultured in serum-free medium. After TFR1 knockdown, ROS accumulation and malignant behaviours were measured. The CCK-8 assay was performed to detect cell viability after TFR1 knockdown and erastin treatment.

### Results

TFR1 expression was upregulated in LIHC tissue and CSCs derived from LIHC cell lines, prompting us to investigate the roles of TFR1 in regulating CSCs. Knockdown of TFR1 expression decreased iron accumulation and inhibited malignant behaviour. Knockdown of TFR1 expression decreased reactive oxygen species (ROS) accumulation induced by erastin treatment and maintained mitochondrial function, indicating that TFR1 is critical in regulating erastin-induced cell death in CSCs. Additionally, knockdown of TFR1 expression decreased sphere formation by decreasing iron accumulation in CSCs, indicating a potential role for TFR1 in maintaining stemness.

**Data Availability Statement:** All relevant data are within the manuscript and its Supporting information files.

**Funding:** This work was supported by Young Scientists Fund of the National Natural Science Foundation of China (Grant No. 81803994), General Program of National Natural Science Foundation of China (Grant No. 81774284), International Cooperation Project of Sichuan Science and Technology Department (Grant No. 2019YFH0152) and Science and Technology Developmental Foundation of Chengdu University of TCM (Grant No. QNXZ2019022) The funders had no role in study design, data collection and analysis, decision to publish, or preparation of the manuscript.

**Competing interests:** The authors have declared that no competing interests exist.

## Conclusion

These findings, which revealed TFR1 as a critical regulator of LIHC CSCs in malignant behaviour and stemness that functions by regulating iron accumulation, may have implications to improve therapeutic approaches.

## Introduction

Cancer stem-like cells (CSCs) are a subpopulation of cancer cells that are characterized by self-renewal and differentiation capacities that are similar to those of normal stem cells [1]. CSCs are tightly associated with tumourigenesis, metastasis and the development of chemotherapy and radiation resistance and may cause tumour relapse after treatment [2]. CSCs derived from hepatocellular carcinoma have been identified by several potential surface markers, including CD24, CD44, and CD133 [3]. They display specific features that enable certain processes, including tumourigenesis, chemoresistance, metastasis and recurrence [4]; thus, investigation of CSCs is needed to develop a potential therapeutic strategy for hepatocellular carcinoma.

Iron is a crucial element in physiological and pathological cell metabolism and biosynthesis and is essential for cell growth and proliferation. Iron homeostasis increases intracellular iron accumulation and availability [5]. Iron uptake is necessary for various vital enzyme functions, including those of mitochondrial enzymes that are essential for the production of energy and reactive oxygen species (ROS), which are responsible for the antioxidant machinery of cells [6]. Iron is also essential in both tumour initiation and tumour progression. In hepatocellular carcinoma, iron overload occurs, leading to ROS formation and mutagenesis [7]. Raggi and colleagues reported that CSCs derived from cholangiocarcinoma expressed increased levels of iron proteins and exhibited high iron accumulation and increased oxidative stress and CSC marker expression compared with parental cells growing as monolayers [8], demonstrating that iron metabolism may also regulate malignancies and CSC stemness in hepatocellular carcinoma.

Transferrin receptor 1 (TFR1) is one of the most crucial proteins for iron uptake and is expressed universally among cell types [9]. Its ligand, transferrin (TF), forms a heterodimer with TFR1 to carry $Fe_2$ for transmembrane transport [10]. TFR1 activity is vital for cancer cells to absorb iron and is deeply involved in tumour onset and progression [11]. In many cancers, TFR1 is significantly dysregulated, and iron uptake is abnormal [12], demonstrating that TFR1 may act as a critical regulator of cancers by affecting iron accumulation. In hepatocellular carcinoma, systemic and intracellular iron homeostasis is altered [6, 13] because of the overexpression of TFR1, indicating the critical role of TFR1 in regulating iron homeostasis [14]. Moreover, TFR1 expression is upregulated in cholangiocarcinoma CSCs, and its activity is associated with increased iron uptake [8]. Knockdown of TFR1 expression decreases iron accumulation and stemness marker expression, indicating a critical role for TFR1 in the stem cell compartment mediated by regulating iron accumulation. However, little is known about how TFR1 affects the malignancy and stemness of CSCs in hepatocellular carcinoma.

This study aimed to determine the expression level and potential roles of TFR1 in the regulation of stemness and malignancy in CSCs derived from hepatocellular carcinoma cells. We show that the presence of TFR1 is essential for iron uptake, malignancy and stemness in CSCs derived from hepatocellular carcinoma. Moreover, TFR1 is critical in regulating ferroptosis-related cell death, indicating its potential as a therapeutic target.

## Material and methods

### Gene expression analysis

Gene Expression Profiling Interactive Analysis (GEPIA, http://gepia.cancer-pku.cn/) was employed as a convenient and accurate online tool based on data from the TCGA and Geno-type-Tissue Expression databases [15]. To analyse the mRNA expression of TFR1, FPN and IRP2, we used GEPIA to compare their expression in LHIC samples and normal samples. Moreover, the Human Protein Atlas (https://www.proteinatlas.org/) database was used for immunohistochemistry (IHC) analysis.

### Cell culture

The hepatocellular carcinoma cell line Huh-7 was obtained from the Japanese Cell Research Bank (catalogue No.: JCRB0403), and SK-HEP-1 was obtained from the American Type Culture Collection (ATCC; Manassas, VA, USA; cat. No.: ATCC®HTB-52™) and stored in our laboratory. These cells were incubated in Dulbecco Modified Eagle Medium-F12 (DMEM/F12; Life Technologies, Grand Island, NY, USA) without 10% foetal bovine serum (FBS; Sigma Chemical Co., St. Louis, MO) supplemented with 2% B-27 (Life Technologies, Grand Island, NY, USA), 20 ng/ml of epidermal growth factor (EGF) and 10 ng/ml of fibroblast growth factor-basic (bFGF; PeproTech, Rocky Hill, NJ, USA). The cells were passaged every 12 days and replated in SFM.

### Serial replating experiments

Cells were replated at clonal density (1,000 cells/well) and cultured in serum-free medium supplemented with 2% B 27, 10 ng/ml of EGF and 20 ng/ml of bFGF. When indicated, the medium was half replaced. After 14 days, the cells were washed with PBS, fixed with 4% paraformaldehyde in PBS, stained with 0.1% crystal violet (Sigma-Aldrich, St. Louis, MO, USA) for 10 min, washed with PBS, and subjected to colony counting. For replating, the same number of cells were plated in SFM. After 14 days, the same procedure was performed.

### Western blotting

Total protein was prepared using RIPA buffer (Thermo Scientific, Waltham, MA, USA) following the manufacturer's instructions. Next, 20 μg of protein was fractionated using Tris-glycine gels and transferred to PVDF membranes. The PVDF membranes were blocked in 5% milk/TBS buffer at room temperature for 60 min and then incubated for 1 h with primary antibodies at a dilution of 1:1000. The following primary antibodies were purchased from Abcam (Cambridge, England): CD24 (cat. No.: ab202073), CD44 (cat. No.: ab18668), CD133 (cat. No.: ab216323), β-actin (cat. No.: ab8226) and TFR1 (cat. No.: ab214039). After washing three times with PBS-T (containing 0.1% Tween-20), the HRP-conjugated secondary antibody (cat. No.: ab6721) was incubated with the PVDF membranes for another 1 h. The blots were developed using Pierce™ ECL Western Blotting Substrate (Thermo Scientific, Waltham, MA, USA) according to the manufacturer's instructions.

### shRNA transfection

shRNAs targeting TRF1 (shTRF1-1 and shTRF1-2) were purchased from Sigma-Aldrich. The relative sequences were as follows: shTRF1-1, 5′- CCCAGCAACAAGACCTTAATA-3′; and shTRF1-2, 5′- CCCTTGATGCACAGTTTGAAA-3′. An shRNA with a scrambled sequence (5′- GAAGCTGCCCACCAGATTG-3′) was used as a negative control (shScrambled). For each transfection, $1\times10^3$ spheres from the second passage (>50 μm in diameter) were

transferred to a 12-well plate, and 0.8 μg of plasmid was mixed with 4 μl of Lipofectamine™ 2000 transfection reagent (Thermo Scientific, Waltham, MA, USA) in 0.5 ml of OptiMEM medium (Thermo Scientific, Waltham, MA, USA); this mixture was then applied to cells for 4 h followed by medium replenishment.

## Cell cycle analysis

Spheres from the second passage were collected and washed 3 times with pre-chilled PBS. Next, 0.25% trypsin was applied to obtain single cells, which were fixed using 1 ml of 75% ice-cold alcohol for 2 h at 4˚C and washed twice with PBS with the addition of 100 μl of RNase A and 400 μl of propidium iodide (PI) (Sigma-Aldrich Chemical Company, St Louis, MO, USA), followed by incubation for 30 min at room temperature. The stained cells were measured using a FACS LSRII flow cytometer (BD Biosciences, Franklin Lakes, NJ, USA), and data were analysed using the ModFit analysis software program (version 4.0; Verity Software House, Inc., Topsham, ME, USA).

## Tumour formation in soft agar

Next, $4 \times 10^3$ single cells from spheres of the second passage were seeded in serum-free RPMI 1640 medium and dissolved in 0.3% low-melting agarose (Sigma-Aldrich, St. Louis, MO, USA) supplemented with 2% B 27, 10 ng/ml of EGF and 20 ng/ml of bFGF. Cells were grown for 14 days, and colonies were visualized with 5% 4-nitro blue tetrazolium chloride staining, for which>50 cells indicated approximately>50 μm in diameter.

## Inductively coupled plasma-mass spectrometry (ICP-MS) analysis

Samples were digested in 70% trace metal basis nitric acid (Sigma-Aldrich, St Louis, MO, USA) and then diluted with double-distilled $H_2O$. The prepared samples were analysed using ICP-MS (Thermo Fisher Scientific, Bremen, Germany) following the manufacturer's instructions. The iron concentration was presented as μg per gram of sample weight. Data are representative of 3 separate experiments.

## Mitochondrial labile iron pool measurement

RPA (Squarix Biotechnology GmbH, Marl, Germany) was used to detect free mitochondrial iron. Cell culture media were removed and replaced with RPA (0.5 μM for 20 min in HBSS (Biotime) at 37˚C), and cells were washed and imaged under an X71 (U-RFL-T) fluorescence microscope (Olympus, Melville, NY).

## CCK-8 assay

Cell viability was determined with the CCK-8 assay. Briefly, for each 96-well plate, 100 spheres from the second passage (>50 μm in diameter) were seeded. After treatment with 0, 0.2 or 0.8 μM erastin for 16 h, 10 μl of freshly prepared CCK-8 solution was added to the cell culture for a 2-h co-incubation. The absorbance was measured at 620 nm.

## Mitochondrial fraction collection

Mitochondria were isolated using a Cell Mitochondria Isolation Kit (Beyotime Jiangsu, China). Briefly, the collected cells were resuspended and homogenized in extraction buffer on ice for 15 min. The cell lysate was then centrifuged at 800 g for 10 min at 4˚C. The pellets were collected as the mitochondrial fraction.

## Statistical analysis

All the results are expressed as the mean ± standard error of the mean (SEM). Differences between two independent groups were evaluated using an unpaired Student's t test, and differences between multiple groups for one or two variables were evaluated using one-way or two-way ANOVA, respectively, with GraphPad Prism 5 software (GraphPad Software, Inc.). Bonferroni's post-hoc test was employed after ANOVA to test all pairwise comparisons. A 2-tailed value of $P < 0.05$ was considered statistically significant.

## Results

### The expression levels of iron metabolism-related genes and proteins are consistently altered in liver hepatocellular carcinoma (LIHC)

We first detected whether iron metabolism-related genes, including TFR1, FPN and IRP2, were altered in LIHC from the TCGA by using the GEPIA online tool (369 LIHC samples and 160 normal samples) (http://gepia.cancer-pku.cn/), which showed that the gene expression levels were upregulated compared with those in normal samples (Fig 1A–1C). Next, we analysed overall survival in patients with high expression of TFR1 but not FPN and IRP2 and found a significantly short overall survival time (TFR1 P = 0.039). Additionally, IHC demonstrated that the TFR1 protein levels in LIHC were upregulated (Fig 1D).

### TFR1 is upregulated in CSCs derived from hepatocellular carcinoma cell lines

To evaluate the role of TFR1 in CSCs derived from hepatocellular carcinoma cell lines, including Huh-7 and SK-HEP-1, we first enriched CSCs by culturing them in serum-free medium. Tumour sphere formation and self-renewal capacity are typical properties of CSCs in vitro [16, 17]. Both cell lines formed unattached tumour spheres and exhibited self-renewal capacity (Fig 2A). Compared with relative parental cells, CSCs presented significantly higher expression levels of CD44 and CD133, two stemness markers [18–20]. Surprisingly, CD24, a stemness marker of HCC [21], was not different between parental cells and CSCs (Fig 2B). Western blotting confirmed that TFR1 was significantly increased in CSCs derived from relative parental cells (Fig 2C). These results indicated that CSCs derived from HCC presented stemness and upregulated TFR1 protein levels.

### TFR1 induces aggressive phenotypes in CSCs derived from human hepatocellular cancer cells

Considering the relatively high endogenous level of TFR1 in both Huh-7- and SK-HEP-1-derived CSCs (Fig 2C), two shRNAs targeting TFR1 mRNA were efficiently introduced into CSCs to knockdown the TFR1 mRNA level (Fig 3A). Compared with the scrambled control group (shScrambled), the TFR1 protein levels in CSCs derived from Huh-7 and SK-HEP-1 cells were significantly downregulated by both shRNAs (Fig 3B), prompting us to transfect mixed shRNAs (shTFR1-1/2). Forty-eight hours after transfection, indicators of malignant behaviours, including cell cycle distribution, invasive capacity and colony formation, were analysed. Consistent with previous reports presenting that TRF1 promotes malignant behaviours in several types of cancers, including breast cancer [22], lung cancer [23], and liver cancer [24], we observed that TFR1 downregulation significantly blocked the cell cycle at G1/G0 phase (Fig 3C). The Transwell assay revealed that the number of transferred cells in the TFR1 knockdown group (26.5±1.7 cells/view for Huh-7; 53.5±1.4 cells/view for SK-HEP-1) was significantly less than that in the shScrambled group (122.5±3.8 cells/view for Huh-7; 153.2±16.5

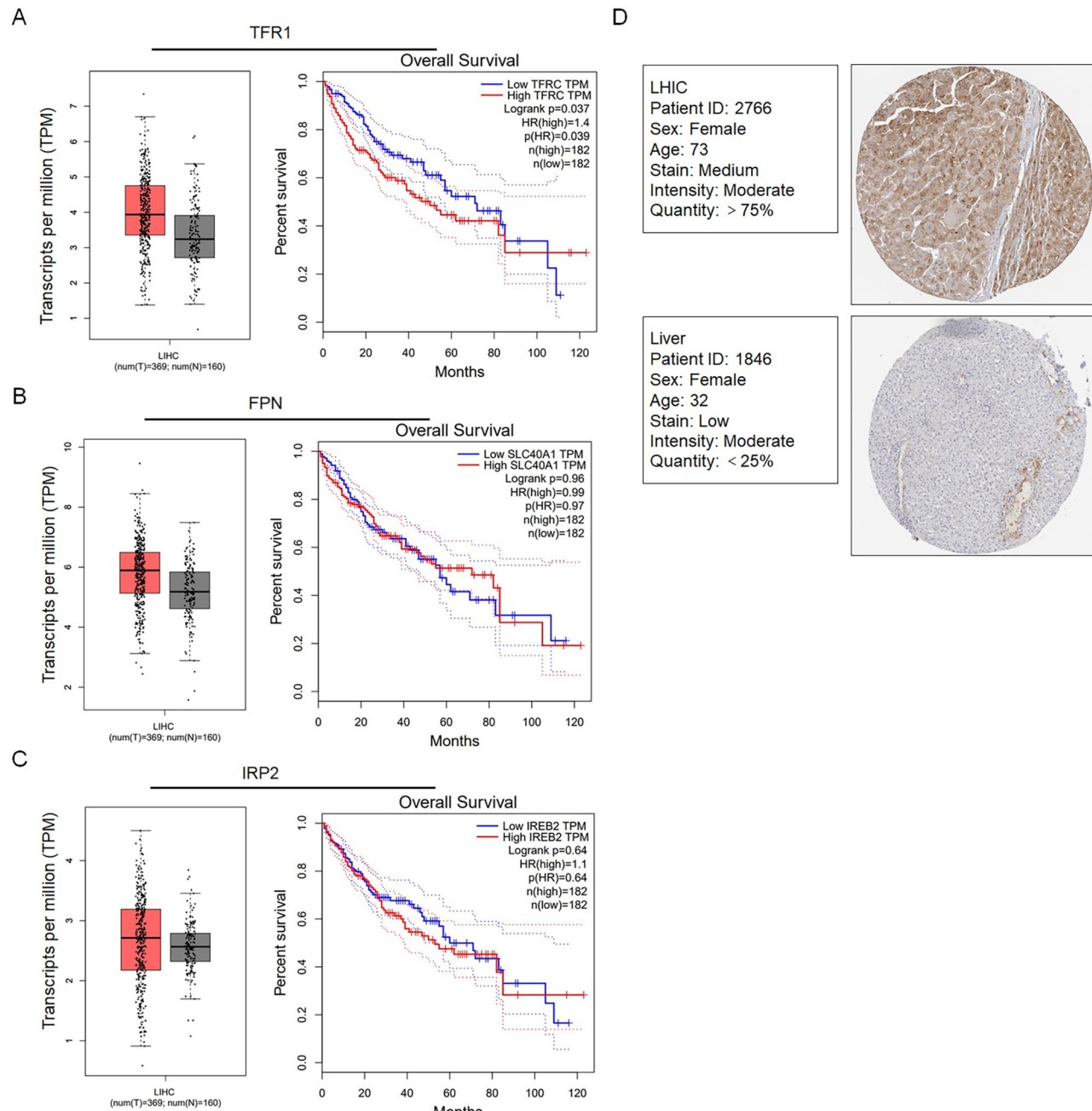

**Fig 1. Expression of iron metabolism-related genes in hepatocarcinoma tissues using the GEPIA platform.** The expression levels of TFR1 (A), FPN (B), and IRP2 (C) in hepatocarcinoma tissues and their relationship with the survival rate of 369 patients were compared with those in normal liver tissues from 160 patients. D. Comparison of TFR1 protein expression between hepatocarcinoma tissues and normal liver tissues from the TCGA CAB005052 dataset in the Human Protein Atlas was performed using immunohistochemistry.

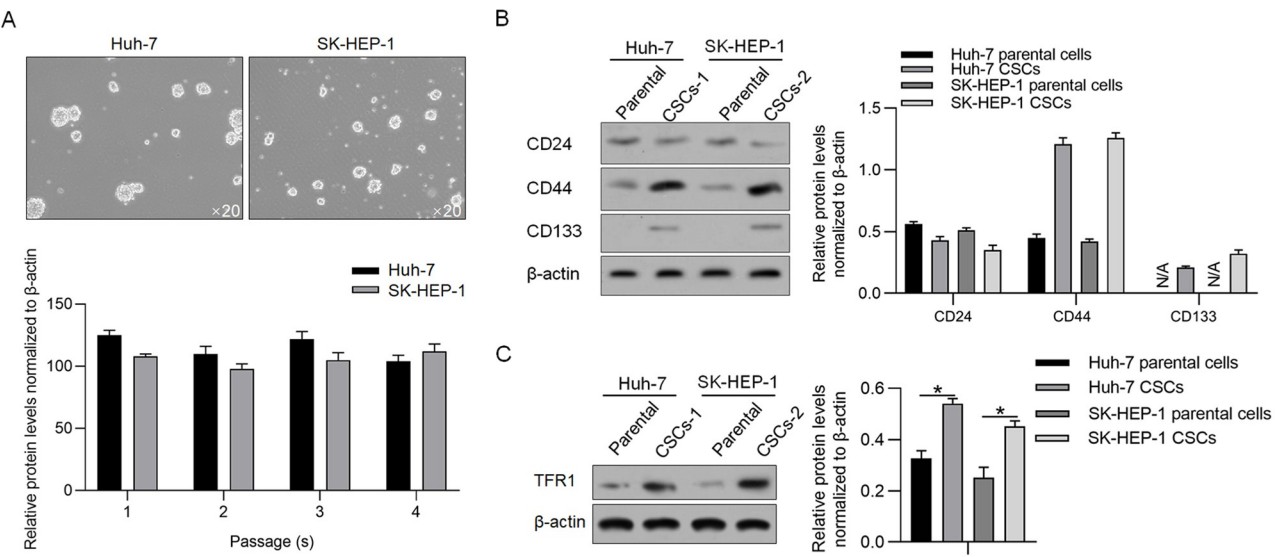

**Fig 2. TFR1 protein levels in CSCs derived from hepatocarcinoma cell lines.** A. Spheres were observed by culturing Huh-7 and SK-HEP-1 cells in serum-free medium. Images were obtained using a 20× objective. B. Stemness markers, including CD24, CD133 and CD44, were measured by performing western blotting. C. TFR1 protein expression was measured in CSCs and compared with that in the parental cells. *P<0.05, vs. parental cells.

cells/view for SK-HEP-1) (Fig 3D). Expectedly, the colony formation ability was also significantly decreased in CSCs derived from Huh-7 or SK-HEP-1 cells (Fig 3E).

## TFR1 regulates iron accumulation

TFR1-induced iron accumulation contributes to the promotion of malignant behaviours in triple-negative breast cancer cells [25]. Thus, we confirmed whether TFR1 regulates iron accumulation in CSCs. Expectedly, after cells were transfected with shTFR1-1/2 for 48 h, the iron concentration in CSCs derived from both Huh-7 and SK-HEP-1 cells was markedly decreased compared with that in shScrambled cells (Fig 4A), indicating that TFR1 is essential for iron uptake. Although CSCs expressed high levels of TFR1, no difference in the iron level was observed in CSCs compared with that in parental cells, likely because of the low iron level in the medium. Thus, CSCs and parental cells were treated with 10 μg/ml of ferric ammonium citrate (FAC, iron source; Sigma-Aldrich). As shown in Fig 4B, the addition of FAC significantly increased iron accumulation in CSCs derived from both Huh-7 and SK-HEP-1 cells, and the knockdown of TFR1 abolished the effect of FAC addition on the iron level. Next, we measured the levels of the mitochondrial labile iron pool after TFR1 knockdown. Knockdown of TFR1 increased the fluorescence signal of rhodamine B4-[(1,10-phenanthroline-5-yl) aminocarbonyl] benzyl ester (RPA) (Fig 4C).

## TFR1 is critical for the erastin-induced cell viability decrease in CSCs

Considering that the expression level of TFR1 is tightly associated with ferroptosis, which is dependent on the presence of iron, we next evaluated the effect of TFR1 on ferroptosis in CSCs. To induce ferroptosis, erastin, a widely used ferroptosis inducer, was employed. Spheres derived from both Huh-7 and SK-HEP-1 cells were dissociated by the addition of erastin (Fig 5A), and a decrease in cell viability was observed (19.3±5.3% in Huh-7 and 27.2±2.9% in SK-HEP-1 after 0.8-μM erastin treatment; Fig 5B). Knockdown of TFR1-1/2 protected spheres

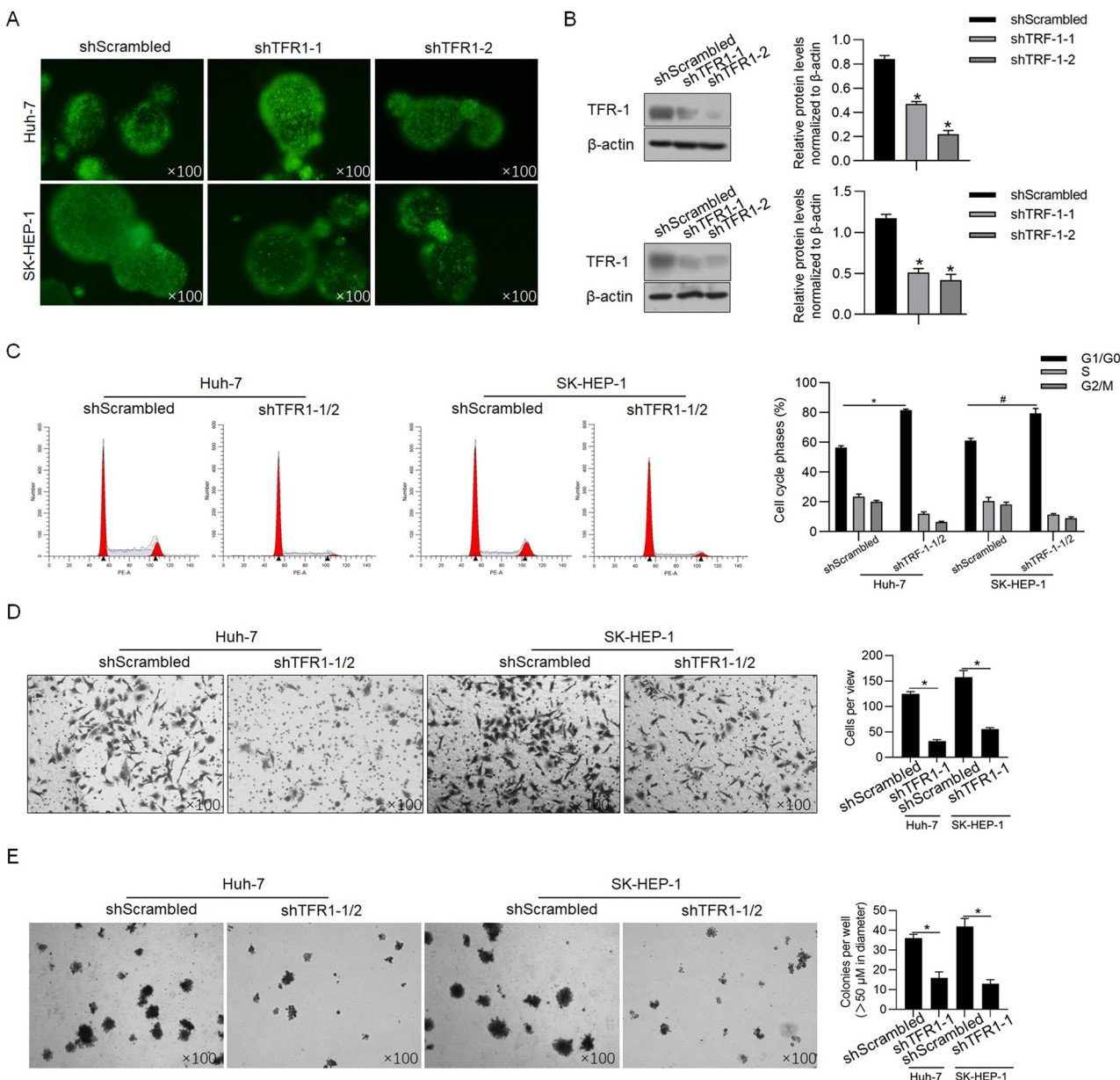

**Fig 3. Effects of TFR1 on the malignant behaviours of CSCs.** A. Fluorescence was observed 48 h after shRNA transfection. Images were obtained using a 100× objective. B. TFR1 knockdown efficacy was measured by performing western blotting. *P<0.05, vs. shScrambled-transfected cells. After TRF1 knockdown in CSCs derived from Huh-7 and SK-HEP-1 cells, malignant behaviours, including cell cycle distribution (C), invasive ability (D) and tumour formation in soft agar (E), were measured. *P<0.05, vs. shScrambled-transfected cells. Images were obtained using a 100× objective.

against erastin in sphere formation and cell viability (49.1±2.5% in Huh-7 and 56.9±1.5% in SK-HEP-1 after 0.8-μM erastin treatment; Fig 5A & 5B). To further confirm whether TFR1 is critical for erastin-induced ferroptosis in CSCs, deferoxamine mesylate (DFO), an iron chelator, was employed as a ferroptosis inhibitor. The addition of DFO reversed the decrease in cell viability induced by erastin, a finding that was similar to the effect of TFR1 knockdown (Fig 5C). Taken together, these data show that TFR1 is critical for erastin-induced ferroptosis in CSCs.

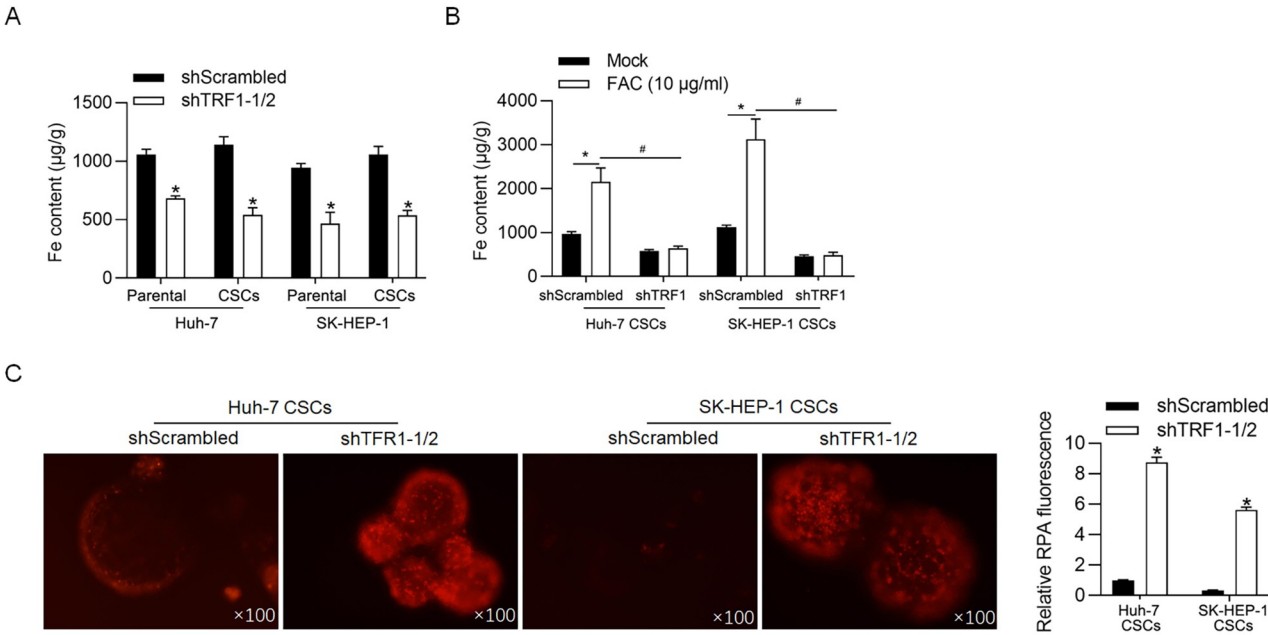

**Fig 4. TFR1 accumulates intracellular iron in CSCs.** A. Intracellular iron levels were measured 48 h later TFR1 knockdown. *P<0.05, vs. shScrambled-transfected cells. B. After the addition of 10 μg/ml of FAC, the intracellular iron levels were measured. *P<0.05, vs. shScrambled-transfected cells. #P<0.05, vs. shScrambled-transfected+FAC cells. C. The level of chelatable intracellular iron was measured with RPA. Images were obtained using a 100× objective. *P<0.05, vs. shScrambled-transfected cells.

## TFR1 regulates mitochondrial function via iron accumulation

Mitochondria are the major sites of intracellular iron distribution, and iron accumulation is tightly related to mitochondrial function [26]. Next, we determined whether TFR1 regulates mitochondrial function after erastin treatment. To explore whether TFR1 affects ROS accumulation, spheres were treated with carboxyl-2', 7'-dichlorofluorescein diacetate (DCFH-DA). The intracellular ROS levels were significantly increased in CSCs after erastin treatment and were significantly decreased by TFR1 knockdown or DFO addition (Fig 6A). Intracellular ROS tightly regulate mitochondrial function, particularly mitochondrial ATP synthesis, by modifying mitochondrial membrane potential homeostasis [27]. Thus, we extracted mitochondria and measured mitochondrial ATP synthesis. Expectedly, knockdown of TFR1 reversed the inhibition of mitochondrial ATP synthesis by erastin, indicating that TFR1 may maintain mitochondrial function under erastin treatment (Fig 6B).

## TFR1 may maintain the stemness of CSCs by regulating iron accumulation

The addition of iron induces the stemness and malignant behaviours of CSCs in human lung cancer cells in an epithelial-to-mesenchymal transition-independent manner without affecting the role of TFR1 [8, 28], indicating that TFR1 may regulate the stemness of CSCs by modifying iron accumulation. Knockdown of TFR1 significantly decreased sphere formation (11.5±7.7% vs 67.5±3.1% in Huh-7 CSCs; 13.6±1.5% vs. 94.5±4.1% in SK-HEP-1 CSCs; p<0.05; Fig 7A). The addition of FAC obviously reversed the effect of the knockdown of TFR1 on sphere formation that was abolished by the addition of DFO (51.2±4.1% vs. 11.5±7.7% in Huh-7 CSCs; 57.4±1.3% vs. 13.6±1.5% in SK-HEP-1 CSCs; p<0.05; Fig 7B). A decrease in CD44 and CD133 also confirmed that the knockdown of TFR1 decreased CSC stemness (Fig 7B). We further

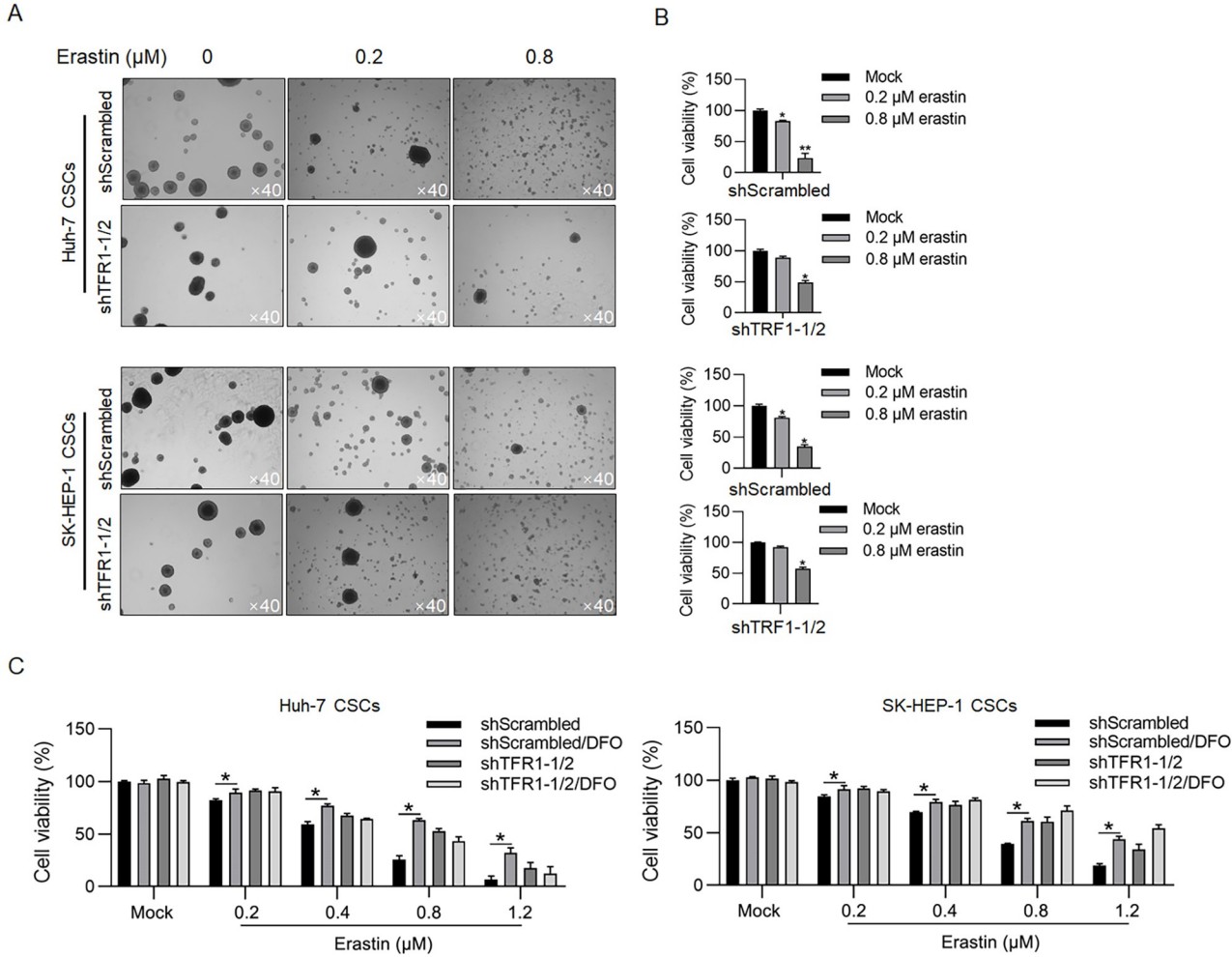

**Fig 5. TFR1 promoted erastin-induced death in CSCs.** A. After TFR1 knockdown, sphere formation under the presence of 0.2 or 0.8 μM erastin was observed. Images were obtained using a 40× objective. B. After TFR1 knockdown, cell viability was measured in the presence of 0.2 or 0.8 μM erastin by performing the CCK-8 assay. *P<0.05, vs. the Mock group; **P<0.01, vs. the Mock group. C. Cell viability was measured in the presence of DFO, a ferroptosis inhibitor. *P<0.05, vs. the shScrambled group.

analysed the cell cycle after TFR1 knockdown using FAC or DFO. The knockdown of TFR1 decreased the number of cells in G1/G0 phase, and the addition of FAC reversed the decrease in G1/G0, which was abolished by the addition of DFO (Fig 7C). This result indicates that TFR1 might also regulate cell proliferation by modifying iron accumulation in CSCs.

## Discussion

Tumour heterogeneity can be used to characterize several different subpopulations, including parental tumour cells (most tumour cells), CSCs (also identified as tumour-initiating cells) and neighbouring cells (including immune cells, cancer-associated fibroblasts and endothelial cells) [29]. These cells comprise the solid tumour and form the tumour microenvironment [30]. Accumulating evidence supports that the presence of CSCs in a subpopulation is essential for tumour growth and progression and that the ability of this subpopulation to induce chemo- and/or radioresistance is the main cause of disease relapse and recurrence. It is crucial

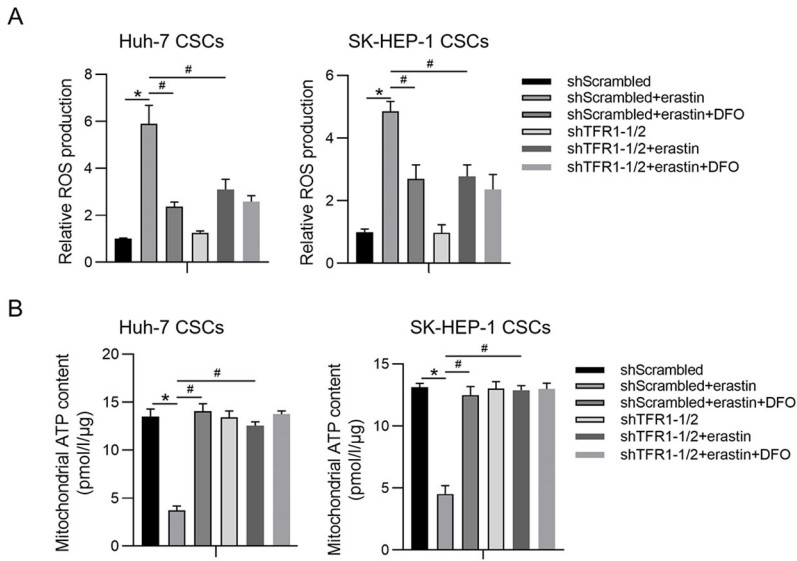

**Fig 6. TFR1 maintains ROS homeostasis by regulating iron accumulation.** A. Intracellular ROS were measured.
*P<0.05, vs. the shScrambled group; #P<0.05, vs. the shScrambled+erastin group. B. Mitochondrial ATP synthesis was
measured. *P<0.05, vs. the shScrambled group; #P<0.05, vs. the shScrambled+erastin group.

to determine the regulatory mechanisms of CSCs to understand their aggressive behaviour
and provide a promising therapeutic strategy for cancer therapy.

The results presented here demonstrate that TFR1 expression is dramatically upregulated
and that TFR1 regulates the iron metabolism balance in CSCs derived from hepatocellular car-
cinoma cells. Specifically, CSCs are characterized by enhanced iron accumulation, which is
reversed by TFR1 knockdown or the addition of DFO. Knockdown of TFR1 expression not
only affects the iron metabolism balance but also decreases CSC malignancy and stemness,
potentially by inducing mitochondrial dysfunction. Increased iron uptake sensitises CSCs to
erastin, a ferroptotic inducer, and leads to increased cell death. Furthermore, our results sug-
gest that CSCs develop an enhanced dependence on iron via upregulated TFR1 expression that
may be targeted by both agents that induce iron-dependent ferroptosis and agents activated by
concentrated iron.

Accumulating evidence has shown that the iron metabolism balance is altered in cancers
and that tumour cells require more iron uptake than their normal or adjacent counterparts to
support several malignant behaviours, including proliferation, migration, invasion and
tumourigenesis [31–35]. Accordingly, most cancer cells exhibit upregulated expression of
TFR1 and reduced levels of FPN, causing intracellular iron accumulation [36]. In this study,
the analysis of TCGA data and survival correlation analysis using GEPIA revealed that TFR1
expression but not FPN or IRP2 expression was upregulated in LIHC tissue compared with
normal liver tissue, prompting us to focus on the potential effect of TFR1 on CSCs derived
from liver cancer cells. Given the well-known inverse correlation between TFR1 and FPN [36],
we surprisingly found that TFR1 was not negatively correlated with FPN, indicating that TFR1
may function in a manner that is dependent on FPN.

In several cancer types, ROS accumulation in tumour cells and the tumour microenviron-
ment promotes carcinogenesis and metastasis [37–39]. Moreover, iron-induced ROS accumu-
lation in normal stem cells and CSCs was also found to augment malignant behaviours, in
part, through enrichment of the CSC stem-like phenotype [28]. Chanvorachote and colleagues

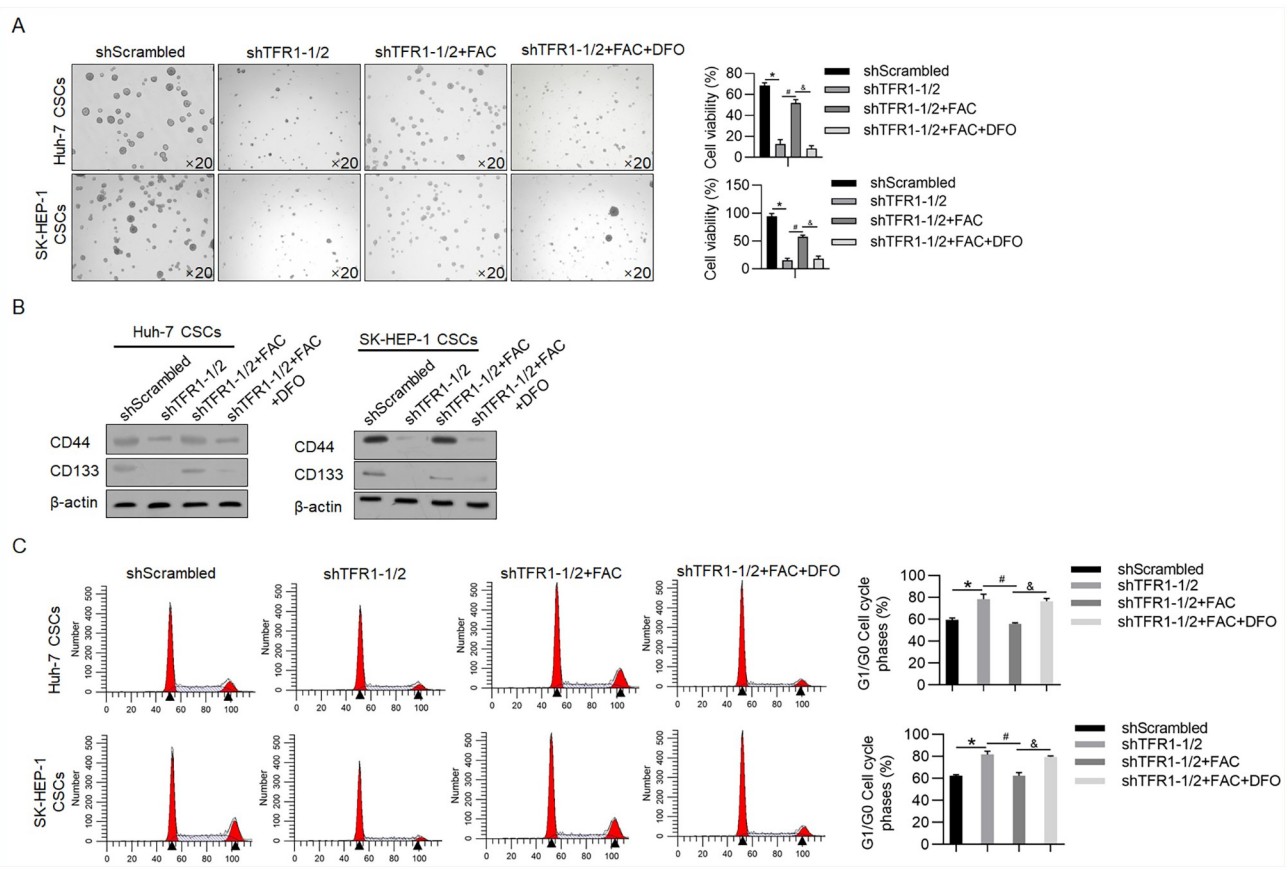

**Fig 7. TFR1 may maintain the stemness and proliferation of CSCs by regulating iron accumulation.** Sphere formation (A), expression of stemness markers (B), and cell cycle (C) analyses were performed after TFR1 knockdown in the presence or absence of FAC and DFO. Images were obtained using a 20× objective. *P<0.05, vs. the shScrambled group. #P<0.05, vs. the shTFR1-1/2 group; &P<0.05, vs. the shTFR1-1/2+FAC group.

reported that in CSCs derived from non-small cell lung tumours, iron exposure and elevated levels of hydroxyl radicals correlated well with an increased CSC subpopulation and promoted malignant behaviours, including sphere formation, proliferation, migration and invasion [28]. They also determined that iron exposure stimulated sex-determining region Y (SRY)-box 9 protein, thus promoting stemness. Consistent with previous findings, we found that iron accumulation induced by TFR1 maintained stemness and promoted malignant behaviours in CSCs derived from hepatocellular carcinoma cells. Moreover, erastin-induced ROS accumulation was significantly decreased by TFR1 knockdown, indicating that TFR1 may be a critical regulator of iron accumulation, thus regulating the malignancy and stemness of CSCs. However, we failed to determine whether TFR1 and FPN are inversely correlated in CSCs, which is worth further investigation.

We also investigated the effect of increased iron accumulation mediated by TFR1 on ferroptosis, which is a characterized programmed cell death process dependent on the presence of iron. As expected, knockdown of TFR1 expression significantly decreased erastin-induced cell death in CSCs, indicating that TFR1 may be an indicator for therapeutic decision-making. Notably, considering the high endogenous level of TFR1, we chose to knock down rather than overexpress TFR1 to study its function. Taken together, our results indicate that a ferroptosis-inducing strategy for the treatment of hepatocellular carcinoma has the potential to address this clinical difficulty.

## Supporting information

**S1 File.**

(ZIP)

## Acknowledgments

The authors would like to thank Mr. Tao Hong for language editing and suggestions for statistical analysis.

## Author Contributions

**Data curation:** Chong Xiao, Hong Liu.

**Formal analysis:** Yuting Wang.

**Funding acquisition:** Ziyi Zhao, Fengming You.

**Investigation:** Chong Xiao, Xi Fu, Yuting Wang, Hong Liu, Yifang Jiang, Ziyi Zhao, Fengming You.

**Methodology:** Chong Xiao, Xi Fu, Ziyi Zhao, Fengming You.

**Project administration:** Ziyi Zhao.

**Resources:** Yuting Wang, Hong Liu.

**Software:** Yuting Wang, Hong Liu, Yifang Jiang.

**Supervision:** Ziyi Zhao, Fengming You.

**Writing – original draft:** Chong Xiao, Xi Fu, Ziyi Zhao, Fengming You.

**Writing – review & editing:** Ziyi Zhao.

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
