## [Decision Letter · Decision Letter 0]

8 Nov 2020

PONE-D-20-32299

Transferrin receptor regulates malignancies and stemness of hepatocellular carcinoma derived cancer stem-like cells via affecting iron accumulation

PLOS ONE

Dear Dr. Zhao,

Thank you for submitting your manuscript to PLOS ONE. After careful consideration, we feel that it has merit but does not fully meet PLOS ONE’s publication criteria as it currently stands. Therefore, we invite you to submit a revised version of the manuscript that addresses the points raised during the review process.

This manuscript is affected by several flaws and mainly by the lack of novelty of the majority of the findings. Regarding the new data they must be better analysed and interpreted. Moreover several other critical points, highlighted by the two referees in their comments, must be taken into consideration and need adequate answers. Lastly English that is considerably substandard, needs to be completely revised by a native English speaker expert in scientific English. The Authors must answer point by point to each criticism raised by the referees in their rebuttal letter, highlighting in red all the amendments made in the text, when submitting their revised version.

We look forward to receiving your revised manuscript.

Kind regards,

Gianpaolo Papaccio, M.D., Ph.D.

Academic Editor

PLOS ONE

Journal Requirements:

2. Please provide additional information about each of the cell lines used in this work, including the source and any quality control testing procedures (authentication, characterisation, and mycoplasma testing). For more information, please see http://journals.plos.org/plosone/s/submission-guidelines#loc-cell-lines.

3. In the Methods section, please provide the source, product number and any lot numbers of the primary antibodies purchased for your study.

4. Please note that PLOS does not permit references to “data not shown.” Authors should provide the relevant data within the manuscript, the Supporting Information files, or in a public repository. If the data are not a core part of the research study being presented, we ask that authors remove any references to these data.

5. To comply with PLOS ONE submission guidelines, in your Methods section, please provide additional information regarding your statistical analyses, including the version of the software used in the analyses. For more information on PLOS ONE's expectations for statistical reporting, please see https://journals.plos.org/plosone/s/submission-guidelines.#loc-statistical-reporting.

6. At this time, we ask that you please provide scale bars on the microscopy images presented in Figure 2, 3, 4, 5 and 7 and refer to the scale bar in the corresponding Figure legend.

7.Thank you for stating the following in the Funding statement Section of your manuscript:

[This work was supported by Young Scientists Fund of the National Natural Science Foundation

of China (Grant No. 81803994), General Program of National Natural Science Foundation of China

(Grant No. 81774284), International Cooperation Project of Sichuan Science and Technology

Department (Grant No. 2019YFH0152) and Science and Technology Developmental Foundation of

Chengdu University of TCM (Grant No. QNXZ2019022）]

 [The funders had no role in study design, data collection and analysis, decision to publish, or preparation of the manuscript.]

8.PLOS ONE now requires that authors provide the original uncropped and unadjusted images underlying all blot or gel results reported in a submission’s figures or Supporting Information files. This policy and the journal’s other requirements for blot/gel reporting and figure preparation are described in detail at https://journals.plos.org/plosone/s/figures#loc-blot-and-gel-reporting-requirements and https://journals.plos.org/plosone/s/figures#loc-preparing-figures-from-image-files. When you submit your revised manuscript, please ensure that your figures adhere fully to these guidelines and provide the original underlying images for all blot or gel data reported in your submission. See the following link for instructions on providing the original image data: https://journals.plos.org/plosone/s/figures#loc-original-images-for-blots-and-gels.

9. PLOS requires an ORCID iD for the corresponding author in Editorial Manager on papers submitted after December 6th, 2016. Please ensure that you have an ORCID iD and that it is validated in Editorial Manager. To do this, go to ‘Update my Information’ (in the upper left-hand corner of the main menu), and click on the Fetch/Validate link next to the ORCID field. This will take you to the ORCID site and allow you to create a new iD or authenticate a pre-existing iD in Editorial Manager. Please see the following video for instructions on linking an ORCID iD to your Editorial Manager account: https://www.youtube.com/watch?v=_xcclfuvtxQ

Reviewers' comments:

Reviewer's Responses to Questions

**Comments to the Author**

1. Is the manuscript technically sound, and do the data support the conclusions?

Reviewer #1: Partly

Reviewer #2: Yes

2. Has the statistical analysis been performed appropriately and rigorously? 

Reviewer #1: I Don't Know

Reviewer #2: Yes

3. Have the authors made all data underlying the findings in their manuscript fully available?

Reviewer #1: Yes

Reviewer #2: Yes

4. Is the manuscript presented in an intelligible fashion and written in standard English?

Reviewer #1: No

Reviewer #2: No

5. Review Comments to the Author

Reviewer #1: In this manuscript the Authors are investigating about the role of transferrin receptor 1 (TFR1) in cancer stem cells (CSCs) from hepatocellular carcinoma. They firstly used patient’s database to show the correlation between TRF1 expression and disease overall survival. Then they showed that TRF1 is overexpressed on CSCs. Knocking down TRF1 results in reduce spheroids formation, cell cycle alteration, reduced colony formation. Moreover, TRF1 knockdown CSCs contain less iron and are resistance to ferroptosis. The work is meaningful in principle, however there are few concerns that need to be addressed as follows:

1. The quality of the English language is poor and the manuscript need a deep revision by a native English speaker.

2. The role of TRF1 in cancer and especially in liver cancer is well documented, the novelty of this work is represented mainly by the role of TRF1 in CSCs. The Authors should point this out more clearly in the manuscript.

3. The Authors state that TRF1 “sensitize” CSCs to erastin; this concept is misleading as TRF1 is normally expressed by CSCs and being TRF1 the major iron transporter is it expected that its knockdown will result in resistance to ferroptosis. All the experiments should be considered as a validation of the role of TRF1 in iron transport in CSCs.

4. One important question regarding CSCs remain unanswered: besides reducing viability does TRF1 knockdown reduce stemness? Even a simple immunoblot with CD44 and CD133 on shTRF1 cells will give at least a preliminary answer.

Reviewer #2: The Authors investigated the expression of transferrin receptor (TFR1) and ferroportin (FPN), two iron importers, and an upstream regulator, iron regulatory protein 2 (IRP2), in liver hepatocellular carcinoma (LIHC) and related CSCs.

The manuscript is very interesting, but there are some points that need to be clarified.

In M&M the authors must indicate what is the passage of spheres used for experiments.

For cell cycle evaluation,the authors must report the software of analysis.

Regarding soft agar, the authors must explain how they detected the colonies. Moreover, how can the authors know that the sphere consists of 50 cells? In my opinion, the authors should evaluate the size.

In figure 2A, the Authors must add the scale bars. The blots showed in Figure 2C must be changed, they are out of focus. Moreover, these results must be better explained in Results Section. It is unclear what is the passage of culture where it has been evaluated the stemness.

In Figure 3B, please change the blots. In Results section, the Authors must make more organic the data regarding cell cycle, indicating, for example, the percentage of cells in each phase.

Also in the case of invasive capacity and colony forming activity, the authors should better describe the results with percentages and differences between shScramble and shTRF.

In Figure 3A, C and D, add the scale bars.

The results showed in figure 4A are unclear. In my opinion, transfected cells showed low levels of iron compared to non transfected cells. In Results section, it is reported there are no differences. Please clarify.

In figure 5A, please add the scale bars. Moreover, cell viability results must be more organic and the authors should stress the differences adding the percentage in text.

In figure 7A, please add the scale bars. Also here, the authors must rewrite the results regarding cell viability and cell cycle, highlighting the differences with percentages.

In addition, the authors should evaluate the stemness in trasfected cells.

In conclusion, the authors should rewrite the results pointing on better the data and the differences among each experimental group. The Discussion Section must be revised and also here, the authors must stress the novelty and limitation of study and not discuss papers already published.

6. PLOS authors have the option to publish the peer review history of their article (what does this mean?). If published, this will include your full peer review and any attached files.

Reviewer #1: No

Reviewer #2: No

---

## [Author Response · Author response to Decision Letter 0]

20 Nov 2020

 Answer: We have confirmed that this manuscript meets the style requirements of PLOS ONE. 

2. Please provide additional information about each of the cell lines used in this work, including the source and any quality control testing procedures (authentication, characterisation, and mycoplasma testing). For more information, please see http://journals.plos.org/plosone/s/submission-guidelines#loc-cell-lines.

 Answer: The description of the cell lines was added to the Materials and methods section as follows:

“Cell culture 

The hepatocellular carcinoma cell line Huh-7 was obtained from the Japanese Cell Research Bank (catalogue no.: JCRB0403), and SK-HEP-1 was obtained from the American Type Culture Collection (ATCC; Manassas, VA, USA; cat. No.: ATCC®HTB-52TM) and stored in our laboratory. These cells were incubated in Dulbecco Modified Eagle Medium-F12 (DMEM/F12; Life Technologies, Grand Island, NY, USA) without 10% foetal bovine serum (FBS; Sigma Chemical Co., St. Louis, MO) supplemented with 2% B-27 (Life Technologies, Grand Island, NY, USA), 20 ng/ml of epidermal growth factor (EGF) and 10 ng/ml of fibroblast growth factor-basic (bFGF; PeproTech, Rocky Hill, NJ, USA). The cells were passaged every 12 days and replated in SFM.”

3. In the Methods section, please provide the source, product number and any lot numbers of the primary antibodies purchased for your study.

 Answer: The catalogue number has been added to the Material and methods section as follows:

“Western blotting 

Total protein was prepared using RIPA buffer (Thermo Scientific, Waltham, MA, USA) following the manufacturer’s instructions. Next, 20 µg of protein was fractionated using Tris-glycine gels and transferred to PVDF membranes. The PVDF membranes were blocked in 5% milk/TBS buffer at room temperature for 60 min and then incubated for 1 h with primary antibodies at a dilution of 1:1000. The following primary antibodies were purchased from Abcam (Cambridge, England): CD24 (cat. No.: ab202073), CD44 (cat. No.: ab18668), CD133 (cat. No.: ab216323), β-actin (cat. No.: ab8226) and TFR1 (cat. No.: ab214039). After washing three times with PBS-T (containing 0.1% Tween-20), the HRP-conjugated secondary antibody (cat. No.: ab6721) was incubated with the PVDF membranes for another 1 h. The blots were developed using PierceTM ECL Western Blotting Substrate (Thermo Scientific, Waltham, MA, USA) according to the manufacturer’s instructions.”

4. Please note that PLOS does not permit references to “data not shown.” Authors should provide the relevant data within the manuscript, the Supporting Information files, or in a public repository. If the data are not a core part of the research study being presented, we ask that authors remove any references to these data.

 Answer: We have modified the text accordingly.

5. To comply with PLOS ONE submission guidelines, in your Methods section, please provide additional information regarding your statistical analyses, including the version of the software used in the analyses. For more information on PLOS ONE's expectations for statistical reporting, please see https://journals.plos.org/plosone/s/submission-guidelines.#loc-statistical-reporting.

Answer: The Statistical analysis section has been modified as follows:

“Statistical analysis 

All the results are expressed as the mean ± standard error of the mean (SEM). Differences between two independent groups were evaluated using an unpaired Student’s t test, and differences between multiple groups for one or two variables were evaluated using one-way or two-way ANOVA, respectively, with GraphPad Prism 5 software (GraphPad Software, Inc.). Bonferroni’s post-hoc test was employed after ANOVA to test all pairwise comparisons. A 2-tailed value of P < 0.05 was considered statistically significant.”

6. At this time, we ask that you please provide scale bars on the microscopy images presented in Figure 2, 3, 4, 5 and 7 and refer to the scale bar in the corresponding Figure legend.

 Answer: We have added this information.

7.Thank you for stating the following in the Funding statement Section of your manuscript:

[This work was supported by Young Scientists Fund of the National Natural Science Foundation

of China (Grant No. 81803994), General Program of National Natural Science Foundation of China

(Grant No. 81774284), International Cooperation Project of Sichuan Science and Technology

Department (Grant No. 2019YFH0152) and Science and Technology Developmental Foundation of

Chengdu University of TCM (Grant No. QNXZ2019022）]

 [The funders had no role in study design, data collection and analysis, decision to publish, or preparation of the manuscript.]

 Answer: We have included this information in the cover letter.

8.PLOS ONE now requires that authors provide the original uncropped and unadjusted images underlying all blot or gel results reported in a submission’s figures or Supporting Information files. This policy and the journal’s other requirements for blot/gel reporting and figure preparation are described in detail at https://journals.plos.org/plosone/s/figures#loc-blot-and-gel-reporting-requirements and https://journals.plos.org/plosone/s/figures#loc-preparing-figures-from-image-files. When you submit your revised manuscript, please ensure that your figures adhere fully to these guidelines and provide the original underlying images for all blot or gel data reported in your submission. See the following link for instructions on providing the original image data: https://journals.plos.org/plosone/s/figures#loc-original-images-for-blots-and-gels.

 Answer: All the original uncropped and unadjusted images were attached as supporting information.

9. PLOS requires an ORCID iD for the corresponding author in Editorial Manager on papers submitted after December 6th, 2016. Please ensure that you have an ORCID iD and that it is validated in Editorial Manager. To do this, go to ‘Update my Information’ (in the upper left-hand corner of the main menu), and click on the Fetch/Validate link next to the ORCID field. This will take you to the ORCID site and allow you to create a new iD or authenticate a pre-existing iD in Editorial Manager. Please see the following video for instructions on linking an ORCID iD to your Editorial Manager account: https://www.youtube.com/watch?v=_xcclfuvtxQ

 Answer: The corresponding author has linked the ORCID ID.

5. Review Comments to the Author

Reviewer #1: In this manuscript the Authors are investigating about the role of transferrin receptor 1 (TFR1) in cancer stem cells (CSCs) from hepatocellular carcinoma. They firstly used patient’s database to show the correlation between TRF1 expression and disease overall survival. Then they showed that TRF1 is overexpressed on CSCs. Knocking down TRF1 results in reduce spheroids formation, cell cycle alteration, reduced colony formation. Moreover, TRF1 knockdown CSCs contain less iron and are resistance to ferroptosis. The work is meaningful in principle, however there are few concerns that need to be addressed as follows:

1. The quality of the English language is poor and the manuscript need a deep revision by a native English speaker.

 Answer: The language has been edited by American Journal Experts, a professional language editing service.

2. The role of TRF1 in cancer and especially in liver cancer is well documented, the novelty of this work is represented mainly by the role of TRF1 in CSCs. The Authors should point this out more clearly in the manuscript.

 Answer: We have included this information in the Introduction section as follows:

Introduction section:

“Transferrin receptor 1 (TFR1) is one of the most crucial proteins for iron uptake and is expressed universally among cell types [9]. Its ligand, transferrin (TF), forms a heterodimer with TFR1 to carry Fe2 for transmembrane transport [10]. TFR1 activity is vital for cancer cells to absorb iron and is deeply involved in tumour onset and progression [11]. In many cancers, TFR1 is significantly dysregulated, and iron uptake is abnormal [12], demonstrating that TFR1 may act as a critical regulator of cancers by affecting iron accumulation. In hepatocellular carcinoma, systemic and intracellular iron homeostasis is altered [6,13] because of the overexpression of TFR1, indicating the critical role of TFR1 in regulating iron homeostasis [14]. Moreover, TFR1 expression is upregulated in cholangiocarcinoma CSCs, and its activity is associated with increased iron uptake [8]. Knockdown of TFR1 expression decreased iron accumulation and stemness marker expression, indicating a critical role for TFR1 in the stem cell compartment mediated by regulating iron accumulation. However, little is known about how TFR1 affects the malignancy and stemness of CSCs in hepatocellular carcinoma.”

3. The Authors state that TRF1 “sensitize” CSCs to erastin; this concept is misleading as TRF1 is normally expressed by CSCs and being TRF1 the major iron transporter is it expected that its knockdown will result in resistance to ferroptosis. All the experiments should be considered as a validation of the role of TRF1 in iron transport in CSCs.

 Answer: Thank you for the suggestions. The word “sensitize” is misleading, and we have modified the description of “sensitivity” throughout the manuscript.

4. One important question regarding CSCs remain unanswered: besides reducing viability does TRF1 knockdown reduce stemness? Even a simple immunoblot with CD44 and CD133 on shTRF1 cells will give at least a preliminary answer.

 Answer: The identification of decreased stemness after TRF1 knockdown was performed as described in figure 7b .

Reviewer #2: The Authors investigated the expression of transferrin receptor (TFR1) and ferroportin (FPN), two iron importers, and an upstream regulator, iron regulatory protein 2 (IRP2), in liver hepatocellular carcinoma (LIHC) and related CSCs.

The manuscript is very interesting, but there are some points that need to be clarified.

In M&M the authors must indicate what is the passage of spheres used for experiments.

 Answer: The passage used for experiments was described as follows:

“CCK-8 assay 

Cell viability was determined with the CCK-8 assay. Briefly, for each 96-well plate, 100 spheres from the second passage (＞50 μm in diameter) were seeded. After treatment with 0, 0.2 or 0.8 μM erastin for 16 h, 10 μl of freshly prepared CCK-8 solution was added to the cell culture for a 2-h co-incubation. The absorbance was measured at 620 nm.”

“Tumour formation in soft agar 

Next, 4×103 single cells from spheres of the second passage were seeded in serum-free RPMI 1640 medium and dissolved in 0.3% low-melting agarose (Sigma-Aldrich, St. Louis, MO, USA), supplemented with 2% B 27, 10 ng/ml of EGF and 20 ng/ml of bFGF. Cells were grown for 14 days, and colonies were visualised with 5% 4-nitro blue tetrazolium chloride staining, for which ＞50 cells indicated approximately ＞50 μm in diameter.”

“shRNA transfection 

shRNAs targeting TRF1 (shTRF1-1 and shTRF1-2) were purchased from Sigma-Aldrich. The relative sequences were as follows: shTRF1-1, 5’- CCCAGCAACAAGACCTTAATA-3’; and shTRF1-2, 5’- CCCTTGATGCACAGTTTGAAA-3’. An shRNA with a scrambled sequence (5’- GAAGCTGCCCACCAGATTG-3’) was used as a negative control (shScrambled). For each transfection, 1×103 spheres from the second passage (＞50 μm in diameter) were transferred to a 12-well plate and 0.8 μg of plasmid was mixed with 4 μl of Lipofectamine™ 2000 transfection reagent (Thermo Scientific, Waltham, MA, USA) in 0.5 ml of OptiMEM medium (Thermo Scientific, Waltham, MA, USA); this mixture was then applied to cells for 4 h followed by medium replenishment.”

“Cell cycle analysis

Spheres from the second passage were collected and washed 3 times with pre-chilled PBS. Next, 0.25% trypsin was applied to obtain single cells, which were fixed using 1 ml of 75% ice-cold alcohol for 2 h at 4℃ and washed twice with PBS with the addition of 100 μl of RNase A and 400 μl of propidium iodide (PI) (Sigma-Aldrich Chemical Company, St Louis, MO, USA), followed by incubation for 30 min at room temperature. The stained cells were measured using a FACS LSRII flow cytometer (BD Biosciences, Franklin Lakes, NJ, USA), and data were analysed using the ModFit analysis software program (version 4.0; Verity Software House, Inc., Topsham, ME, USA).”

For cell cycle evaluation, the authors must report the software of analysis.

 Answer: The software used to analyse the cell cycle was described as follows:

“Cell cycle analysis

Spheres from the second passage were collected and washed 3 times with pre-chilled PBS. Next, 0.25% trypsin was applied to obtain single cells, which were fixed using 1 ml of 75% ice-cold alcohol for 2 h at 4℃ and washed twice with PBS with the addition of 100 μl of RNase A and 400 μl of propidium iodide (PI) (Sigma-Aldrich Chemical Company, St Louis, MO, USA), followed by incubation for 30 min at room temperature. The stained cells were measured using a FACS LSRII flow cytometer (BD Biosciences, Franklin Lakes, NJ, USA), and data were analysed using the ModFit analysis software program (version 4.0; Verity Software House, Inc., Topsham, ME, USA).”

Regarding soft agar, the authors must explain how they detected the colonies. Moreover, how can the authors know that the sphere consists of 50 cells? In my opinion, the authors should evaluate the size.

 Answer: The staining of colonies was described, and colonies approximately ＞50 µm in diameter were counted.

“Tumour formation in soft agar 

Next, 4×103 single cells from spheres of the second passage were seeded in serum-free RPMI 1640 medium and dissolved in 0.3% low-melting agarose (Sigma-Aldrich, St. Louis, MO, USA), supplemented with 2% B 27, 10 ng/ml of EGF and 20 ng/ml of bFGF. Cells were grown for 14 days, and colonies were visualised with 5% 4-nitro blue tetrazolium chloride staining, for which ＞50 cells indicated approximately ＞50 μm in diameter.”

In figure 2A, the Authors must add the scale bars. The blots showed in Figure 2C must be changed, they are out of focus. Moreover, these results must be better explained in Results Section. It is unclear what is the passage of culture where it has been evaluated the stemness.

 Answer: Figure 2 has been modified and attached, and the results were modified specifically as follows:

“TFR1 is upregulated in CSCs derived from hepatocellular carcinoma cell lines 

To evaluate the role of TFR1 in CSCs derived from hepatocellular carcinoma cell lines, including Huh-7 and SK-HEP-1, we first enriched CSCs by culturing them in serum-free medium. Tumour sphere formation and self-renewal capacity are typical properties of CSCs in vitro [16,17]. Both cell lines formed unattached tumour spheres and exhibited self-renewal capacity (Figure 2A). Compared with relative parental cells, CSCs presented significantly higher expression levels of CD44 and CD133, two stemness markers [18,19,20]. Surprisingly, CD24, a stemness marker of HCC [21], was not different between parental cells and CSCs (Figure 2B). Western blotting confirmed that TFR1 was significantly increased in CSCs derived from relative parental cells (Figure 2C). These results indicated that CSCs derived from HCC presented stemness and upregulated TFR1 protein levels.” 

In Figure 3B, please change the blots. In Results section, the Authors must make more organic the data regarding cell cycle, indicating, for example, the percentage of cells in each phase.

Answer: Figure 3B has been modified.

Also in the case of invasive capacity and colony forming activity, the authors should better describe the results with percentages and differences between shScramble and shTRF.

Answer: The results were further described as follows: 

“Considering the relatively high endogenous level of TFR1 in both Huh-7- and SK-HEP-1-derived CSCs (Figure 2C), two shRNAs targeting TFR1 mRNA were efficiently introduced into CSCs to knockdown the TFR1 mRNA level (Figure 3A). Compared with the scrambled control group (shScrambled), the TFR1 protein levels in CSCs derived from Huh-7 and SK-HEP-1 cells were significantly downregulated by both shRNAs (Figure 3B), prompting us to transfect mixed shRNAs (shTFR1-1/2). Forty-eight hours after transfection, indicators of malignant behaviours, including cell cycle distribution, invasive capacity and colony formation, were analysed. Consistent with previous reports presenting that TRF1 promotes malignant behaviours in several types of cancers, including breast cancer [22], lung cancer [23], and liver cancer [24], we observed that TFR1downregulation significantly blocked the cell cycle at G1/G0 phase (Figure 3C). The Transwell assay revealed that the number of transferred cells in the TFR1 knockdown group (26.5±1.7 cells/view for Huh-7; 53.5±1.4 cells/view for SK-HEP-1) was significantly less than that in the shScrambled group (122.5±3.8 cells/view for Huh-7; 153.2±16.5 cells/view for SK-HEP-1) (Figure 3D). Expectedly, the colony formation ability was also significantly decreased in CSCs derived from Huh-7 or SK-HEP-1 cells (Figure 3E).”

In Figure 3A, C and D, add the scale bars.

Answer: The amplification is shown in the figures.

The results showed in figure 4A are unclear. In my opinion, transfected cells showed low levels of iron compared to non transfected cells. In Results section, it is reported there are no differences. Please clarify.

Answer: The results of figure 4A were further described as follows:

“TFR1-induced iron accumulation contributes to the promotion of malignant behaviours in triple-negative breast cancer cells [25]. Thus, we confirmed whether TFR1 regulates iron accumulation in CSCs. Expectedly, after cells were transfected with shTFR1-1/2 for 48 h, the iron concentration in CSCs derived from both Huh-7 and SK-HEP-1 cells was markedly decreased compared with that in shScrambled cells (Figure 4A), indicating that TFR1 is essential for iron uptake. Although CSCs expressed high levels of TFR1, no difference in the iron level was observed in CSCs compared with that in parental cells, likely because of the low iron level in the medium. Thus, CSCs and parental cells were treated with 10 μg/ml of ferric ammonium citrate (FAC, iron source; Sigma-Aldrich). As shown in figure 4B, the addition of FAC significantly increased iron accumulation in CSCs derived from both Huh-7 and SK-HEP-1 cells, and the knockdown of TFR1 abolished the effect of FAC addition on the iron level. Next, we measured the levels of the mitochondrial labile iron pool after TFR1 knockdown. Knockdown of TFR1 increased the fluorescence signal of rhodamine B4-[(1,10-phenanthroline-5-yl) aminocarbonyl] benzyl ester (RPA) (figure 4C).”

In figure 5A, please add the scale bars. Moreover, cell viability results must be more organic and the authors should stress the differences adding the percentage in text.

Answer: The amplification of each image was added, and the results were further described as follows:

“Considering that the expression level of TFR1 is tightly associated with ferroptosis, which is dependent on the presence of iron, we next evaluated the effect of TFR1 on ferroptosis in CSCs. To induce ferroptosis, erastin, a widely used ferroptosis inducer, was employed. Spheres derived from both Huh-7 and SK-HEP-1 cells were dissociated by the addition of erastin (figure 5A), and a decrease in cell viability was observed (19.3±5.3% in Huh-7 and 27.2±2.9% in SK-HEP-1 after 0.8-μM erastin treatment; figure 5B). Knockdown of TFR1-1/2 protected spheres against erastin in sphere formation and cell viability (49.1±2.5% in Huh-7 and 56.9±1.5% in SK-HEP-1 after 0.8-μM erastin treatment; figure 5A&B). To further confirm whether TFR1 is critical for erastin-induced ferroptosis in CSCs, deferoxamine mesylate (DFO), an iron chelator, was employed as a ferroptosis inhibitor. The addition of DFO reversed the decrease in cell viability induced by erastin, a finding that was similar to the effect of TFR1 knockdown (figure 5C). Taken together, these data show that TFR1 is critical for erastin-induced ferroptosis in CSCs.” 

In figure 7A, please add the scale bars. Also here, the authors must rewrite the results regarding cell viability and cell cycle, highlighting the differences with percentages.

Answer: The amplification of each image was added, and the differences with percentages were described as follows:

“The addition of iron induces the stemness and malignant behaviours of CSCs in human lung cancer cells in an epithelial-to-mesenchymal transition-independent manner without affecting the role of TFR1 [28,29], indicated that TFR1 may regulate the stemness of CSCs by modifying iron accumulation. Knockdown of TFR1 significantly decreased sphere formation (11.5±7.7% vs 67.5±3.1% in Huh-7 CSCs; 13.6±1.5% vs. 94.5±4.1% in SK-HEP-1 CSCs; p＜0.05; figure 7a). The addition of FAC obviously reversed the effect of the knockdown of TFR1 on sphere formation that was abolished by the addition of DFO (51.2±4.1% vs. 11.5±7.7% in Huh-7 CSCs; 57.4±1.3% vs. 13.6±1.5% in SK-HEP-1 CSCs; p＜0.05; figure 7a). A decrease in CD44 and CD133 also confirmed that the knockdown of TFR1 decreased CSC stemness (figure 7b). We further analysed the cell cycle after TFR1 knockdown using FAC or DFO. The knockdown of TFR1 decreased the number of cells in G1/G0 phase, and the addition of FAC reversed the decrease in G1/G0, which was abolished by the addition of DFO (figure 7c). This result indicates that TFR1 might also regulate cell proliferation by modifying iron accumulation in CSCs.”

In addition, the authors should evaluate the stemness in trasfected cells.

Answer: The stemness was evaluated in figure 7b.

In conclusion, the authors should rewrite the results pointing on better the data and the differences among each experimental group. The Discussion Section must be revised and also here, the authors must stress the novelty and limitation of study and not discuss papers already published.

Answer: The Discussion section has been modified.

---

## [Decision Letter · Decision Letter 1]

27 Nov 2020

Transferrin receptor regulates malignancies and the stemness of hepatocellular carcinoma-derived cancer stem-like cells by affecting iron accumulation

PONE-D-20-32299R1

Dear Dr. Zhao,

We’re pleased to inform you that your manuscript has been judged scientifically suitable for publication and will be formally accepted for publication once it meets all outstanding technical requirements.

Kind regards,

Gianpaolo Papaccio, M.D., Ph.D.

Academic Editor

PLOS ONE

Additional Editor Comments (optional):

Reviewers' comments:

Reviewer's Responses to Questions

**Comments to the Author**

1. If the authors have adequately addressed your comments raised in a previous round of review and you feel that this manuscript is now acceptable for publication, you may indicate that here to bypass the “Comments to the Author” section, enter your conflict of interest statement in the “Confidential to Editor” section, and submit your "Accept" recommendation.

Reviewer #1: All comments have been addressed

Reviewer #2: All comments have been addressed

2. Is the manuscript technically sound, and do the data support the conclusions?

Reviewer #1: Yes

Reviewer #2: Yes

3. Has the statistical analysis been performed appropriately and rigorously? 

Reviewer #1: I Don't Know

Reviewer #2: Yes

4. Have the authors made all data underlying the findings in their manuscript fully available?

Reviewer #1: Yes

Reviewer #2: Yes

5. Is the manuscript presented in an intelligible fashion and written in standard English?

Reviewer #1: Yes

Reviewer #2: Yes

6. Review Comments to the Author

Reviewer #1: The Authors have addressed all the concerns raised by this reviewer. The Manuscript is now greatly improved.

Reviewer #2: (No Response)

7. PLOS authors have the option to publish the peer review history of their article (what does this mean?). If published, this will include your full peer review and any attached files.

Reviewer #1: No

Reviewer #2: No

---

## [Editor Report · Acceptance letter]

10 Dec 2020

PONE-D-20-32299R1 

Transferrin receptor regulates malignancies and the stemness of hepatocellular carcinoma-derived cancer stem-like cells by affecting iron accumulation 

Dear Dr. Zhao:

I'm pleased to inform you that your manuscript has been deemed suitable for publication in PLOS ONE. Congratulations! Your manuscript is now with our production department. 

Kind regards, 

on behalf of

Prof. Gianpaolo Papaccio 

Academic Editor

PLOS ONE